# How does the farmland management rights mortgage loan affect food security: Based on the evidence of major grain producing areas in China

Xiangyu Dong[1], Xiangbo Cheng[2]*, Shuiling Huang[2]*, Xuran Li[2], Yiru Wang[2]

**1** Zhejiang A&F University, School of Humanities and Law, Hangzhou, China, **2** Zhejiang A&F University, School of Economics and Management, Hangzhou, China

* cxb@stu.zafu.edu.cn (XC); 20200080@zafu.edu.cn (SH)

## Abstract

The people are the foundation of the nation, and grain is vital for their survival. Food security is crucial for national stability and well-being. This study utilizes panel data from 1,234 counties in China's primary grain-producing regions, covering the years 2010 to 2021, to conduct a theoretical analysis and empirical examination of the impact of farmland management rights mortgage loans (FMRML) on food security, employing a difference-in-differences (DID) model. The empirical results indicate that the pilot policy significantly enhances food security in major grain-producing areas. Moreover, the conclusions remain robust following parallel trends and placebo tests. Mechanism analysis reveals that the policy can promote food security by increasing rural mechanization and boosting financial support in rural areas. Heterogeneity analysis demonstrates that factors such as regional resource endowments, economic development levels, and farmland tenure systems can lead to varying effects of the policy on food security. This research provides valuable references and insights for global practices aimed at enhancing food security through policies enabling the mortgage of farmland management rights.

## Introduction

In this new development stage, amidst a complex and changing international environment, regional conflicts, and resource and environmental constraints in agricultural production, it is particularly crucial to solidify the foundation of food security comprehensively [1,2]. Following the imperative to synergistically promote carbon reduction, pollution control, afforestation, and growth, and by the era's requirements of ecological priority, conservation and intensification, and green, low-carbon development, safeguarding national food security is of utmost importance. Despite advancements in science and technology enabling increased agricultural productivity from natural resources, food security remains a global challenge [3,4]. The FAO released the State of Food Security and Nutrition in the World report, forecasting the most severe challenges in history. Between 690 million to 783 million people worldwide faced hunger in 2022. Hunger levels continue to rise in low-income countries and

**Data availability statement:** All relevant data are within the manuscript and its Supporting Information files.

**Funding:** This study was funded by the General Research Project of Zhejiang Provincial Department of Education (Grant numbers Y202353113). The funders had no role in study design, data collection and analysis, decision to publish, or preparation of the manuscript.

**Competing interests:** The authors have declared that no competing interests exist.

regions, with the possibility of severely food-insecure populations reaching 943 million by 2025, highlighting ongoing global concerns regarding food insecurity. China, as a developing country with an enormous population, had a per capita grain availability of 483 kilograms as of 2021, surpassing the internationally recognized food security line of 400 kilograms. The self-sufficiency rate of cereals exceeds 95%, with rice and wheat achieving rates of over 100%[1]. While China is relatively secure regarding basic food security metrics, long-term trends reveal ongoing challenges. On the one hand, there are issues such as limited arable land, scarce water resources, rural labor migration leading to declining workforce quality, and a fragile foundation for grain cultivation. On the other hand, high production costs and declining profitability in grain farming diminish farmers' incentive to grow crops, thereby restricting overall grain output capacity and posing further threats to food security. Policy-driven agricultural collateral loans can secure grain cultivation, accelerate land transfer, promote rural income growth, and thereby enhance farmers' enthusiasm for cultivating crops, ultimately safeguarding food security [5–7].

Rural finance for agriculture serves as an effective means to promote agricultural production development, ensuring food security, and fostering sustainable agricultural development [8,9]. In recent years, to deepen rural financial reform and enhance the performance of credit support for agriculture, rural credit products have exhibited a diversified nature. Particularly, leveraging farmland to obtain funds has effectively addressed the issue of insufficient collateral among farmers, playing a crucial catalytic role in overcoming the persistent challenges of "difficulty and high cost of financing" for farmers [10–12]. To accelerate the dissemination and deepening of the FMRML model, the central government has continuously issued a series of policy documents, thereby establishing a comprehensive policy support system. Since 2014, the document titled "Opinions on Guiding the Orderly Transfer of Farmland Management Rights to Develop Moderate Scale Agriculture" has clarified the principle of "separation of the three rights," laying a legal foundation for the transfer of farmland management rights. Subsequently, in 2015, the State Council issued the "Guiding Opinions on the Pilot Projects for Mortgaging Farmland Management Rights and Farmers' Housing Property Rights," which officially initiated the pilot exploration of FMRML financing. Following this, in 2016, the People's Bank of China promulgated the "Interim Measures for the Pilot of Mortgaging Rural Contracted Farmland Management Rights," marking the formal establishment of the FMRML system at the institutional level. This development has greatly stimulated the financial vitality of farmland as an operating asset, expanding farmers' financing channels and injecting new momentum into agricultural production efficiency. This initiative has rapidly taken root in 232 counties nationwide, becoming a vivid practice of rural financial innovation. The implementation of this series of policies has not only facilitated the effective transfer of farmland and the steady increase of farmers' income but has also played a significant role in maintaining national food security and agricultural ecological safety. This paper aims to fill the current research gap regarding the relationship between farmers and farmland, particularly in the context of financial support for agriculture. By analyzing the supportive effects of national financial policies, it explores how to optimize the structure of farmland use, enhance production efficiency, and stabilize farmers' incomes. Importantly, the policy of FMRML is expected to alleviate the tensions between policies and land resources, fostering close collaboration among the government, financial sectors, and farmers. This cooperation will collectively strengthen the defense against food insecurity and guide farmers to focus more on improving food quality and safety, thereby promoting healthier and more sustainable agricultural development [13].

The issue of food security has become a significant concern for both the government and academia. Currently, there is a wealth of existing research on the factors influencing food

security, including agricultural endowments, climate change, and trade services [14–16]. Specifically, from the perspective of agricultural endowments, land serves as a core element, and its optimal allocation is crucial for ensuring food security [17]. Developing countries have implemented land transfer policies to optimize resource allocation and enhance food security [18]. From the perspective of climate change, the uncertainties posed by climate change exacerbate the impacts on food production systems, increasing the vulnerability of food supply [19,20]. From a trade perspective, trade liberalization promotes global agricultural cooperation and resource sharing while also igniting profound discussions regarding food self-sufficiency and reliance on imports [21–23]. Upon reviewing the literature related to food security, it is evident that existing research has extensively explored the various factors influencing food security. Additionally, studies have also focused on the impact of land policies and finance on food security. However, as a reform policy that combines farmland factors with financial instruments, research on the impact of FMRML on food security has not been thoroughly conducted.

Current research analyzing the income effects of FMRML involves or implicitly suggests their significance for food security. Besley found that FMRML effectively alleviates rural credit constraints, increases agricultural capital investment, enhances agricultural efficiency, and promotes the improvement of farmers' income levels, thereby empowering the development of food security [24]. Stupen proposed through analytical research that FMRML is a highly effective approach to enhancing agricultural productivity, improving agricultural efficiency, and increasing farmers' income [25]. Yang et al. demonstrated that participation in FMRML encourages farmers to invest more in the agricultural sector, improving agricultural productivity and indirectly ensuring food security [26]. Additionally, literature has explored the relationship between finance and food security. Lin et al. found that rural finance supports agricultural output growth by meeting the financial needs of agricultural production through credit assistance, enabling farmers to increase agricultural investment, which has a significant positive impact on food security [27]. Liu and Ren, using a two-way fixed effects model, discovered that digital inclusive finance can overcome temporal and spatial limitations as well as regional discrimination [28]. By enhancing financial accessibility, expanding the scope of financing, and increasing financial supply, promotes the formation of high-quality agricultural development models across regions, ensuring food security while facilitating a low-carbon transformation in food production. Similarly, Li et al. analyzed the impact of farmers' financial literacy on factor substitution and crop structure adjustment behaviors on food production using data from the 2021 China Land Economic Survey from both theoretical and empirical perspectives [29]. Existing literature has conducted a relatively systematic exploration of FMRML; however, there is limited literature directly examining the actual effects of FMRML on county-level food security, making it difficult to provide rich and robust empirical evidence for the effectiveness of FMRML policy implementation.

In summary, this study focuses on the field of land use policy and confronts the issue of food security, placing FMRML and food security directly in the same analytical framework. The direct impact of the policy on food security is verified from both theoretical and empirical perspectives, and the mechanisms underlying this effect are explored. Consequently, the marginal contributions of this paper can be articulated in four key areas. First, the uniqueness of the study: This research may represent the first systematic analysis of how the FMRML influences food security based on extensive county-level data, particularly during the transformation of rural economic structures and the modernization of agriculture. By integrating the liquidity of farmland management rights with the stability of food production, this study fills a significant gap in the existing literature and provides new research perspectives in relevant fields. Such research not only aids in understanding how farmland management rights

function as financial instruments to promote agricultural development but also reveals their potential impacts on food security. Second, the practical significance of the study: This paper offers effective recommendations for policymakers aimed at optimizing the policy of FMRML to enhance both agricultural production and food security. In light of the current financing challenges in the transfer of farmland management rights, it is suggested that the government improve relevant policies by providing more flexible loan conditions and safeguards, thereby reducing farmers' financing costs and promoting their motivation and production capacity. These measures are expected to contribute to stable growth in food production, thereby enhancing overall food security. Third, theoretical and empirical contributions: By exploring the mechanisms through which the policy of FMRML affects food security, this study expands the existing theoretical framework and proposes corresponding hypotheses, offering new ideas and perspectives for theoretical research. Utilizing survey data from major grain-producing areas in China, the understanding of the mechanisms by which the policy influences food security is deepened, providing empirical evidence for subsequent research and validating specific hypotheses. Additionally, the study examines the variations in responses to the policy across different regions, economic levels, and farmland tenure systems, providing robust support for practices in related fields. Fourth, the potential impact of the research: The findings of this study are expected to have a profound effect on the policy of FMRML, facilitating the improvement of food security measures and promoting the coordinated development of agricultural modernization and rural economies. As global food security challenges become increasingly severe, China faces the significant challenge of enhancing agricultural production efficiency and self-sufficiency in food supply. By proposing a more effective policy of FMRML, this research will assist in guiding governments and enterprises in the formulation of better policies and strategies, ensuring that farmers remain motivated and capable of supplying food, ultimately promoting the sustainable and efficient development of Chinese agriculture.

## Policy background

The household contract responsibility system has facilitated the rapid development of agriculture and rural areas in China. However, the separation of ownership, contracting rights, and operational rights for rural land has impeded the effective utilization of land assets for financial purposes. Consequently, farmers have been unable to use their land as collateral to secure bank loans. Data indicates that, despite the substantial inventory of resource assets in rural regions of China, property income has historically contributed less than 4% to farmers' overall income. This has weakened agricultural operations and led to a noticeable degree of fallow arable land [30].

The first clear guidance on mortgage loans for farmland was provided by the No. 1 Central Document of 2009, along with the subsequent joint issuance of the "Guiding Opinions on Further Strengthening Credit Structure Adjustment to Promote Stable and Rapid Economic Development" by the People's Bank of China and the China Banking Regulatory Commission. This document mentioned that "certain areas may explore the establishment of mortgage loans for land use rights," thus officially initiating the pilot phase of farmland mortgage loans. Encouraged by central policies, provinces meeting specific conditions soon formulated corresponding pilot measures and actively undertook pilot activities. As practical explorations continued, significant legal adjustments finally began to unfold. At the end of 2015, the central government selected 232 counties (cities, districts) as pilot areas. In early 2016, pilot projects were launched in these designated regions. Additionally, the General Office of the State Council published the "Opinions on Improving the Division of Ownership, Contract Rights, and Management Rights of Rural Land," which clearly distinguished between land contract

rights and management rights. This policy implemented a parallel system regarding ownership, contract rights, and management rights. The entities managing the land were required to obtain written consent from the contracting farmers or their authorized representatives before transferring management rights or establishing a mortgage by laws and regulations, with written record-keeping required by the farmers' collective. Furthermore, the opinions outlined an intention to achieve complete coverage of rural property transaction markets in agricultural counties to better provide services such as information dissemination, property transactions, legal consultation, rights evaluation, and mortgage financing for both parties involved in land transfers.

As the pilot projects yielded significant results, the legislative process for the separation of rights accelerated. In 2018, the seventh meeting of the 13th National People's Congress Standing Committee passed a decision to amend the Farmland Contracting Law, marking the first clear legal definition of farmland management rights. In 2020, the third session of the 13th National People's Congress adopted the Civil Code. By the relevant directives from the Central Committee and the amended Farmland Contracting Law, Chapter 11 of the property section of the Civil Code provided detailed regulations specifically concerning "farmland contract management rights." Consequently, the concept of the "separation of three rights" in farmland has been endowed with substantial legal significance, making it a timely moment for the nationwide implementation of FMRML.

## Mechanism and research hypothesis

### The impact of FMRML on the food security: comprehensive effect

Ensuring food security relies fundamentally on support in terms of people, land, and finances [31,32]. For a long time, Chinese farmers have been unable to utilize their farmland as effective collateral to secure production funds, severely limiting their enthusiasm for agricultural production. This situation has led to continual rural labor outmigration and widespread farmland abandonment [33–35]. The introduction of policies allowing for the mortgage of farmland use rights has officially recognized the collateral attributes of these rights at the policy level [36,37]. Such policies transform farmland assets into efficient capital, providing essential financial support to agricultural entities, easing credit constraints, and enhancing credit accessibility [38]. Upon obtaining loans, farmers can partially alleviate financial scarcity issues, thereby stimulating the output of agricultural production capital and labor inputs. This approach leverages financial alleviation effects, thereby benefiting the enhancement of comprehensive food production capabilities and ensuring food security [39,40]. Furthermore, the policy of FMRML contributes to activating farmland and human capital elements, thereby bolstering food supply capabilities. Agricultural production and management necessitate financial resources and funding support. On the one hand, mortgaging farmland use rights facilitates farmland transfer, activating underutilized or inefficiently used arable farmland resources, thereby optimizing their utilization and expanding the scale of grain production operations, ultimately increasing total food production [41]. On the other hand, increased financial support promotes investment in labor force training, enhancing workforce professionalism and quality, optimizing labor deployment, increasing the efficiency of agricultural production, and consequently augmenting food output and quality [42].

### The impact of FMRML on the food security: mechanism analysis

In recent years, the policy benefits of FMRML for food security have begun to manifest. This is primarily achieved through advancing mechanization levels at the county level and developing financial support levels. Regarding the mechanization level transmission mechanism,

agricultural production of grains necessitates support from mechanical technology, and the impact of the policy of FMRML on mechanization levels cannot be ignored [43]. On the one hand, insufficient mechanization levels at the county level increase risks in agricultural production management, potentially hindering the implementation of FMRML and undermining their deep-level application in the grain production industry, thus weakening the policy's impact on grain cultivation. On the other hand, the policy of FMRML provides farmers with a new avenue to obtain funds for purchasing agricultural machinery and equipment, thereby enhancing mechanization levels. Its development further increases agricultural production efficiency, strengthens agricultural resilience against natural disasters, optimizes agricultural industry structure, and effectively safeguards both the quantity and quality of grain production [9].

Regarding the transmission mechanism of financial support levels, on the one hand, the policy of FMRML uses farmers' farmland management rights as collateral, endowing them with the ability to secure loans and finance, thereby increasing the asset value of farmland. This helps in reducing credit costs and risks for county-level financial institutions, increasing rural financial supply, further activating local financial markets, and enhancing financial support levels. On the other hand, the elevation of local financial support levels can provide more formal financial support for agricultural operators, continuously channeling financial resources into the grain production sector, thereby incentivizing inputs in agricultural elements and positively promoting food security[44].

## The impact of FMRML on the food security: heterogeneity analysis

The formulation of the policy of FMRML must take into account regional resource endowments, economic development levels, and farmland tenure systems. These policies should be tailored to local conditions and implemented progressively, exploring diverse approaches and regulatory mechanisms for the implementation of FMRML in practice.

On one hand, geographical location may play a heterogeneous role in the impact of pilot policies on food security. The main grain-producing areas can be categorized into northern and southern regions. The northern region is primarily characterized by plains, where the scale of agricultural operations is generally larger. In contrast, the southern region, dominated by hilly and mountainous terrain, faces challenges due to fragmented farmland, which hinders large-scale agricultural operations. This fragmentation exacerbates issues such as abandoned farmland and a shift in cropping structure away from food production [45]. On the other hand, there are significant disparities in economic development levels between regions. In China, the eastern region exhibits higher levels of economic development and greater government investment, which often leads to a more proactive implementation of pilot policies [46]. Effective policy support facilitates the rational allocation of farmland and labor resources, significantly enhancing agricultural production efficiency and capacity, thereby promoting the scaling up of food production at the county level and ensuring food security. Additionally, differences in farmland tenure systems can also affect the effectiveness of pilot policies [47]. In areas with a high cropping index, where farmland use efficiency is optimized, farmers can significantly increase grain yields through enhanced investment and expanded production scale. Consequently, the policy of FMRML can more effectively contribute to food security in such regions. Therefore, it is likely that the policy of FMRML will yield heterogeneous policy effects across different regions.

Based on the above theoretical analysis, the impact mechanism of the policy of FMRML on food security is shown in Fig 1, and the following research hypotheses are proposed:

Hypothesis 1 (H1). The policy of FMRML can promote the food security.

Hypothesis 2 (H2). The policy of FMRML can promote the food security by increasing the level of mechanization and financial support.

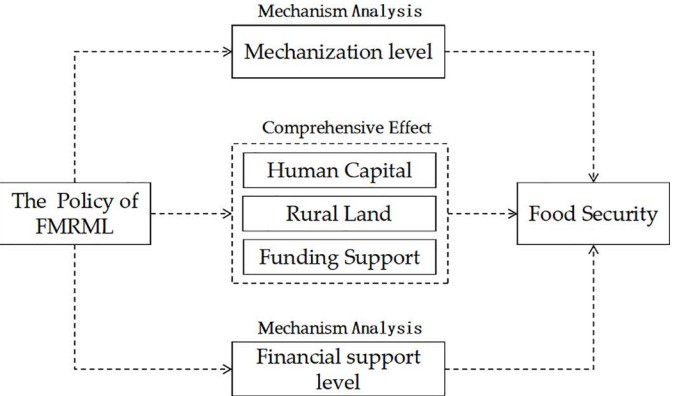

**Fig 1. Theoretical mechanism.**

Hypothesis 3 (H3). The impact of the policy of FMRML on the food security varies significantly according to regional resource endowments, economic development levels, and farmland tenure systems.

## Research method

### Identification strategy

The DID model has been widely applied in the field of policy evaluation in recent years. It identifies the net effect of policies intuitively by comparing differences in outcomes before and after policy interventions between individuals affected and those unaffected by the policy. Additionally, it can be easily estimated using ordinary least squares, offering the advantages of intuitive understanding and simplicity [48]. Before applying the DID model, two types of variables need to be constructed. On the one hand, there are policy variables: in this study, counties participating in the pilot program for FMRML in major grain-producing areas are assigned a value of 1, while non-participating counties are assigned a value of 0. After excluding counties with severe data deficiencies or administrative changes, the final sample includes 124 pilot counties and 1110 non-pilot counties. On the other hand, there are time variables: this study assigns a value of 1 after the implementation of the policy, and a value of 0 before its implementation. Specifically, the pilot policy was initiated in 2016, with the period from 2010 to 2015 considered as the pre-implementation phase, and from 2016 to 2021 as the post-implementation phase. Finally, the study constructs interaction terms between policy and time variables, introducing them as core explanatory variables into the DID model to examine the net effects of the policy of FMRML on food security.

### Baseline regression models

This paper employs a DID model to assess the impact of the policy of FMRML on food security in major grain-producing areas. Pilot counties implementing the policy of FMRML as the treatment group, while other counties serve as the control group. The structure of the model is as follows:

$$Y_{it} = \alpha_0 + \alpha_1 DID_{it} + \alpha_2 X_{it} + \gamma_t + \mu_i + \varepsilon_{it} \tag{1}$$

In Equation (1), $Y_{it}$ as the explained variable, representing the food security in county $i$ of the major grain-producing area in year $t$. $DID_{it}$ serves as the key explanatory variable, calculated

as $DID_{it} = post_t \times treat_i$, where $post_t$ is a time dummy variable taking the value 1 if the observation year $t$ is in or after the policy implementation year, and 0 otherwise; $treat_i$ is a dummy variable indicating whether county $i$ belongs to the pilot area, taking the value 1 if it does, and 0 otherwise. The double-difference term $DID_{it}$ indicates whether county $i$ is part of the pilot program for FMRML in year $t$. The coefficient $\alpha_1$ is the focal point of interest in this study. $X_{it}$ denotes control variables, $\alpha_2$ represents their coefficients, $\alpha_0$ is a constant term, $\gamma_t$ denotes year fixed effects, $\mu i$ represents region fixed effects, and $\varepsilon_{it}$ denotes the error term.

## Mechanism test model

To examine the pathways and mechanisms through which the policy of FMRML affects food security in major grain-producing areas, this study constructs the following model, drawing upon the mediation analysis framework proposed by Baron and Kenny [49]:

$$M_{it} = \beta_0 + \beta_1 DID_{it} + \beta_2 X_{it} + \gamma_t + \mu_i + \varepsilon_{it} \tag{2}$$

$$Y_{it} = \varphi_0 + \varphi_1 DID_{it} + \varphi_2 M_{it} + \varphi_3 X_{it} + \gamma_t + \mu_i + \varepsilon_{it} \tag{3}$$

In equations (2) and (3), $M_{it}$ represents the mediating variables, encompassing levels of mechanization and financial support. The meanings of other symbols remain consistent with equation (1).

## Variables and data

**Variable selection. Explained variable**. The explained variable in this study is food security, measured by total grain production at the county level, following past research approaches, with the data log transformed [50].

**Core Explanatory variables.** The core explanatory variable of this study is the policy of FMRML. The publication of the list of pilot counties for FMRML provides an appropriate quasi-natural experimental scenario for this research. Thus, the list of pilot counties chosen for FMRML serves as the foundation for the construction of the key explanatory variable in this research. Specifically, if a county was designated as a pilot county in the observed year, it is assigned a value of 1; otherwise, it is assigned a value of 0.

**Mediation variables**. The policy of FMRML may impact food security through two pathways: enhancing levels of mechanization and financial support. Therefore, this study incorporates two mediating variables, mechanization level, and financial support level, for mechanistic analysis. Following Guan et al., agricultural machinery power at the county level in the current year represents the mechanization level and is logged for analysis [51]. Similarly, following the approach of Si, year-end balances of various loans from financial institutions are used to represent the financial support level at the county level and are also logged for analysis [52].

**Control Variables**. Based on earlier studies, this analysis incorporates the following control variables to examine the overall impact of implementing the FMRML policy on food security in major grain-producing areas [53–55]: agricultural development level, where the ratio of the value added of the primary industry to regional GDP signifies agricultural development level; industrial structure, represented by the ratio of the value added of the secondary industry to regional GDP; government intervention, measured by the logarithm of local general public budget expenditures; farming scale, where the logarithm of crop sown area represents county-level farming scale; population density, represented by the ratio of the county's total

year-end population to the area of the county's administrative area; precipitation, measured by the annual average precipitation for each county; and temperature, assessed by the annual average temperature for each county.

## Data sources and descriptive statistics

This study uses panel data from 1234 counties in China's major grain-producing areas spanning from 2010 to 2021 to empirically analyze the effect of the policy of FMRML on food security. The indicator data primarily originate from the China County (City) Social and Economic Statistical Yearbook, China County Statistical Yearbook (County and City Volume), provincial statistical yearbooks, and county statistical bulletins. Where data were missing, linear interpolation was used for completion. Table 1 displays the results of a descriptive statistical analysis of the variables.

## Results

### Baseline regression results

The baseline regression results of the FMRML policy's effect on food security are shown in Table 2. The core explanatory variables' coefficients are all considerably positive, as shown in Column (1), indicating that the policy's advancement has a notable beneficial impact on food security in these regions. In Column (2), after incorporating control variables such as agricultural development level, industrial structure, government intervention, farming scale, population density, precipitation, and temperature, the regression results show a coefficient of 0.062 for Landloan, significant at the 1% level. This demonstrates that regions implementing the policy of FMRML promote food security relative to those that do not, thus supporting Hypothesis H1. Additionally, the regression results for the control variables indicate that agricultural development level, industrial structure, and farming scale have a considerable beneficial impact on food security. Meanwhile, government intervention, population density, and temperature exert certain inhibitory effects on the development of food security.

Possible explanations suggest that the FMRML policy provides financial support to farmers, enabling them to make essential investments, such as purchasing seeds, fertilizers, and agricultural machinery, thereby enhancing production efficiency. Simultaneously, improvements in agricultural development levels and optimization of industrial structures indicate better resource allocation and technological application, which contribute to increased yield

**Table 1. Variable definition and descriptive statistics.**

| Variable Name | Variable Abbreviation | Observations | Mean | Std. | Min | Max |
|---|---|---|---|---|---|---|
| Food security | Lnsafe | 14808 | 12.240 | 1.417 | 3.296 | 15.202 |
| Farmland Management rights mortgage policy | Landloan | 14808 | 0.051 | 0.219 | 0 | 1 |
| Mechanization level | Lnmech | 14808 | 3.473 | 1.247 | 0 | 6.136 |
| Financial support level | Lnfina | 14808 | 12.573 | 3.793 | 0 | 17.732 |
| Agricultural development level | First | 14808 | 0.178 | 0.118 | 0 | 0.755 |
| Industrial structure | Str | 14808 | 0.435 | 0.151 | 0.013 | 1.158 |
| Government intervention | Lngov | 14808 | 12.498 | 0.666 | 8.579 | 15.587 |
| Farming scale | Lncrop | 14808 | 2.643 | 2.174 | -1.640 | 6.382 |
| Population density | Pop | 14808 | 0.056 | 0.141 | 0.001 | 4.184 |
| Precipitation | Rain | 14808 | 0.003 | 0.001 | 0 | 0.009 |
| Temperature | Tem | 14808 | 13.287 | 4.858 | -3.809 | 21.653 |

**Table 2. Baseline Regression Results.**

| Variable | (1) | (2) |
|---|---|---|
| | Lnsafe | |
| Landloan | 0.070*** | 0.062*** |
| | (0.022) | (0.022) |
| First | | 0.307** |
| | | (0.147) |
| Str | | 0.184* |
| | | (0.111) |
| Lngov | | -0.064** |
| | | (0.031) |
| Lncrop | | 0.099*** |
| | | (0.026) |
| Pop | | -1.962* |
| | | (1.061) |
| Rain | | 3.243 |
| | | (4.337) |
| Tem | | -0.024*** |
| | | (0.008) |
| Constant | 12.234*** | 12.975*** |
| | (0.008) | (0.370) |
| Time effects | YES | YES |
| Regional effects | YES | YES |
| R² | 0.005 | 0.037 |
| N | 14808 | 14808 |

Note:

***,

**and

*denote significance at the 1%, 5%, and 10% levels, respectively, and the figures in brackets represent the standard errors.

and quality. In contrast, government interventions may create policy uncertainty, negatively impacting farmers' decision-making and productivity motivation. Additionally, high population density may intensify competition for farmland, leading to strained arable resources, while temperature fluctuations directly affect crop growth conditions, thus impacting food production security.

## Parallel trend test

Fig 2 reports the estimation results of the parallel trends test. The results indicate that the estimated coefficients of the Landloan in the periods before the implementation of the policy of FMRML are not statistically significant, thus failing to reject the null hypothesis of a coefficient of 0. This suggests that the double difference model meets the precondition that the treatment and control groups exhibit parallel trends before policy implementation. In terms of dynamic effects, the regression coefficients of the Landloan become significant starting from the first period after policy implementation, and their impact gradually strengthens over time. This fact also indicates that there is a lag in the impact of the policy on food security in the major grain-producing areas, requiring some time from policy implementation to effectiveness.

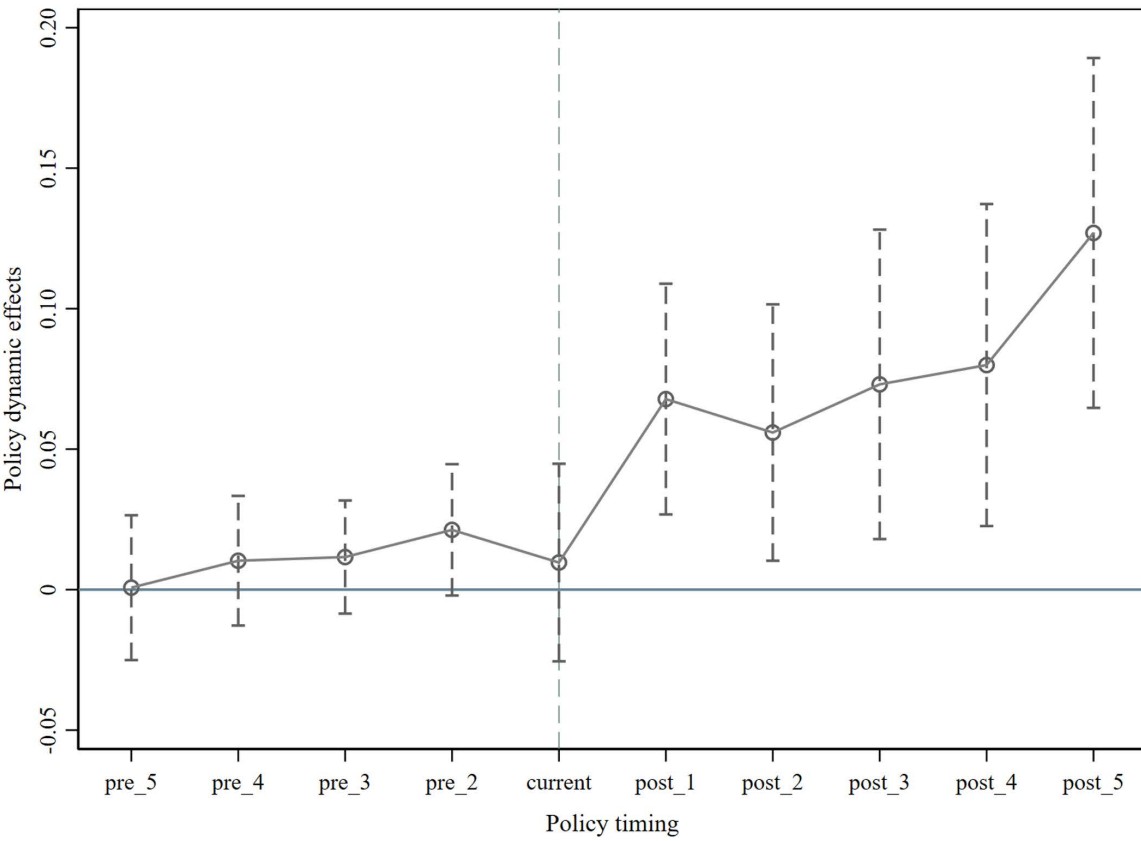

**Fig 2. The dynamic effect.**

## Robustness check

Other unobservable random factors might influence estimation outcomes, such as unobserved regional variables. This study employs a randomized controlled trial approach to re-estimate Model (1) and derive parameter estimates for key explanatory variables. This process is repeated 500 times, followed by kernel density plot creation. As depicted in Fig 3, the solid line represents the probability density of simulated coefficient estimates, conforming to a normal distribution overall, indicating random assignment between treatment and control groups. The coefficient estimate for Landloan in the baseline regression is 0.062, falling outside the distribution of simulated coefficient estimates. This suggests that the implementation of the policy on food security has passed placebo tests, with unobservable random factors failing to influence the estimation results, thereby ensuring unbiased estimation of core explanatory variable coefficients in this study. This further underscores the significant promotional effect of FMRML on food security in major agricultural regions.

## PSM-DID analysis

To address potential endogeneity issues arising from non-random policy implementation and to mitigate selection bias due to systemic errors, this study employs the PSM-DID method to reevaluate the effects of the policy of FMRML. Agricultural development level, industry structure, government intervention, farming scale, population density, precipitation, and

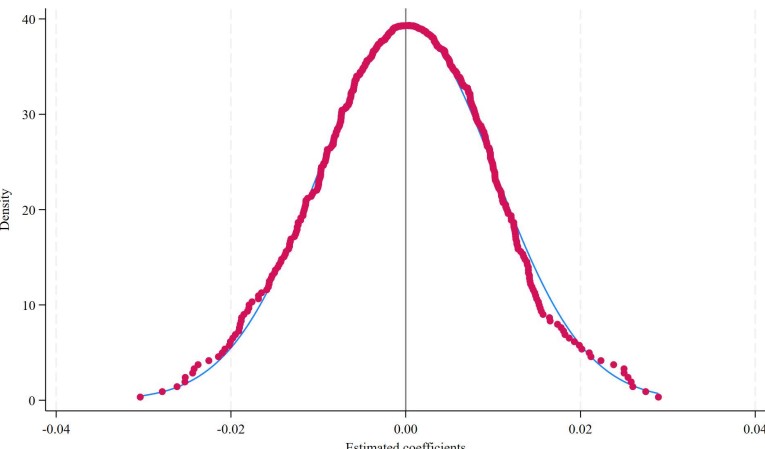

**Fig 3. Placebo test results.**

temperature (defined specifically as control variables) are selected as matching variables to minimize significant differences between the treatment and control groups before the pilot phase of FMRML. Comparing differences before and after propensity score matching of control variables reveals that most control variables exhibit reduced discrepancies post-matching, thereby achieving more precise matching between treatment and control groups and enhancing the accuracy of policy effect estimates (Fig 4). Based on the new sample, double-difference regression estimates are conducted again, with Table 3 (2) presenting the results of the PSM-DID model incorporating bidirectional fixed effects and control variables. The estimates are significant at the 5% level, reaffirming the policy's positive impact on food security in major grain-producing areas.

## Replacement of explained variables

In the aforementioned baseline regression model, the dependent variable used is the logarithm of total food production. Furthermore, this study employs per capita total food production as an alternative variable in another round of baseline regression model estimation. The distinction lies in the former considering the scale and capability of overall food production, while

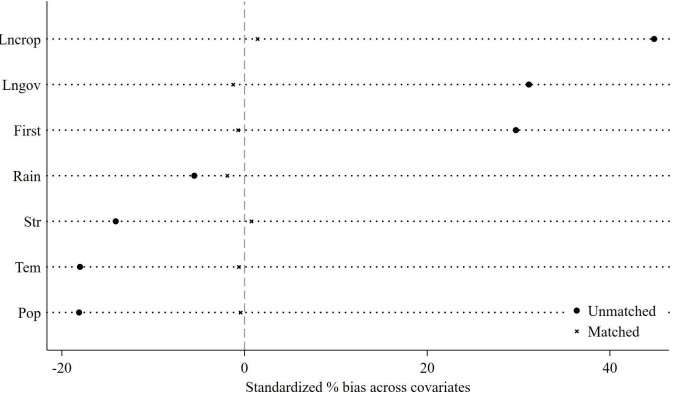

**Fig 4. Variable differences before and after PSM.**

**Table 3. Robustness analysis results.**

| Variable | (1) Baseline regression results | (2) PSM-DID | (3) Replacement of Explained Variables |
|---|---|---|---|
| Landloan | 0.062*** | 0.030** | 0.072* |
| | (0.022) | (0.012) | (0.038) |
| Constant | 12.975*** | 12.832** | 0.762** |
| | (0.370) | (0.244) | (0.330) |
| Control variables | YES | YES | YES |
| Time effects | YES | YES | YES |
| Regional effects | YES | YES | YES |
| R² | 0.037 | 0.024 | 0.030 |
| N | 14808 | 14808 | 14808 |

the latter focuses more on individual food supply conditions. Regression findings are shown in Table 3 (3) with the substituted dependent variable, accounting for region-fixed effects, time-fixed effects, and a chosen subset of control variables. The findings show that the regression coefficient has grown and is statistically positively associated at the 10% level, indicating that the implementation of the policy continues to significantly promote food security.

## Mechanism analysis

Mechanization levels and financial support are crucial factors influencing food security in major grain-producing areas. Despite previous findings demonstrating that the implementation of the policy of FMRML significantly enhances food security, it is imperative to further investigate whether such policies can promote food security through intensified mechanization and financial support levels. The regression results of the mechanism analysis are presented in Table 4. Column (1) of Table 4 presents the overall impact of the policy of FMRML on food security, consistent with baseline regression results. Column (2) examines the empirical evidence of these policies on mechanization levels, indicating a positive impact at the 1% significance level. Column (3) explores the combined impact of the policy of FMRML and mechanization levels on food security, both significant at the 5% and 1% levels respectively, highlighting a significant mediating effect of mechanization levels on food security in major agricultural regions facilitated by these policies. Similarly, Column (4) examines the empirical evidence of these policies on financial support levels, significant at the 5% level. Column (5) demonstrates the combined impact of the policy of FMRML and financial support levels on food security, significant at the 1% level, demonstrating a significant mediating influence of enhanced financial support on food security in major grain-producing areas promoted by these policies. Furthermore, the overall effect of the policy of FMRML on food security in major grain-producing areas is 0.062. When introducing variables for mechanization levels and financial support levels separately, the direct effects of these policies on food security are 0.056 and 0.060, respectively. The total effect exceeds the direct effects, suggesting that mechanization levels and financial support levels partially mediate the promotion of food security by the policy of FMRML. In conclusion, Hypothesis H2 is validated.

## Heterogeneity analysis

**Geographic heterogeneity.** Significant regional disparities exist in China's grain production environment. Based on the varying natural resource endowments of major grain-producing provinces, these provinces are categorized into northern and southern grain-producing regions. Using benchmark regression models for estimation, this study further

**Table 4. Test Results of Influence Mechanism.**

| | (1)<br>Lnsafe | (2)<br>Lnmech | (3)<br>Lnsafe | (4)<br>Lnfina | (5)<br>Lnsafe |
|---|---|---|---|---|---|
| Landloan | 0.062*** | 0.129*** | 0.056** | 0.061** | 0.060*** |
| | (0.022) | (0.026) | (0.022) | (0.030) | (0.022) |
| Lnmech | | | 0.049*** | | |
| | | | (0.018) | | |
| Lnsafe | | | | | 0.035*** |
| | | | | | (0.009) |
| Control variables | YES | YES | YES | YES | YES |
| Time effects | YES | YES | YES | YES | YES |
| Regional effects | YES | YES | YES | YES | YES |
| Constant | 12.975*** | 3.270*** | 12.815*** | 8.712*** | 12.669*** |
| | (0.370) | (0.494) | (0.381) | (1.008) | (0.375) |
| $R^2$ | 0.037 | 0.030 | 0.041 | 0.489 | 0.042 |
| N | 14808 | 14808 | 14808 | 14808 | 14808 |

examines the differential impacts of the policy of FMRML on food security across these regions. The southern grain-producing region comprises six provinces: Hubei, Anhui, Hunan, Jiangsu, Sichuan, and Jiangxi; while the northern grain-producing region includes seven provinces: Hebei, Henan, Jilin, Heilongjiang, Liaoning, Shandong, and Inner Mongolia. The findings are presented in columns (1) and (2) in Table 5. The coefficient measuring the impact of the FMRML policy on food security is 0.079 for the northern region, which is statistically significant at the 5% level. In contrast, for the southern region, the coefficient is 0.032 and not statistically significant. This disparity may stem from the relatively abundant farmland resources in the northern region, potentially underutilized compared to the higher utilization rates in the southern region. Thus, the policy of FMRML may more effectively unleash the farmland potential in the northern grain-producing region, thereby enhancing farmland use efficiency and grain production.

**Heterogeneity in levels of economic development.** Different stages of regional development may result in varying impacts of the policy of FMRML on food security. To address this, the overall sample is segmented into the more developed eastern regions and the less developed central and western regions. Columns (3) and (4) in Table 5 indicate that in

**Table 5. Heterogeneity Analysis.**

| Variable | Geographic location | | The level of economic development | |
|---|---|---|---|---|
| | (1)northern grain producing regions | (2)southern grain producing regions | (3)eastern regions | (4)central and western regions |
| Landloan | 0.079** | 0.032 | 0.101* | 0.031 |
| | (0.033) | (0.023) | (0.061) | (0.019) |
| Control Variables | YES | YES | YES | YES |
| Time effects | YES | YES | YES | YES |
| Regional effects | YES | YES | YES | YES |
| Constant | 12.902*** | 12.735*** | 14.515*** | 12.105*** |
| | (0.625) | (0.385) | (0.831) | (0.336) |
| $R^2$ | 0.106 | 0.052 | 0.158 | 0.029 |
| N | 7740 | 7068 | 4680 | 10128 |

the eastern regions, the policy of FMRML significantly promotes food security. Conversely, in the central and western regions, the effects of policy implementation are not significant. This disparity may be attributed to higher levels of agricultural productivity and farmer income in the eastern regions, providing a robust economic foundation for the implementation of the policy. In contrast, weaker infrastructure, underdeveloped financial markets, and inadequate policy support in the central and western regions hinder farmers' access to sufficient loan capital for agricultural production, thereby affecting the overall effectiveness of the policy on food security. Thus, this study verifies Hypothesis 3, asserting that the impact of the policy of FMRML on food security in major grain-producing areas varies significantly depending on geographical location and economic development level.

**Heterogeneity in cropland systems.** Differences exist in the farming systems adopted across various regions, primarily reflected in the variations of the multiple cropping index. The multiple cropping index refers to the ratio of cultivated area to arable farmland area, indicating the frequency of crop planting within a single year per unit of arable farmland. To examine whether the impact of the policy on food security is influenced by the farming systems, sample cities were categorized into high and low multiple cropping index regions based on the median of the multiple cropping index in the sample regions. The grouped regression results are presented in columns (5) and (6) in Table 5. The policy of FMRML has a stronger positive effect on food security in regions with a low multiple cropping index. The analysis suggests that regions with a low multiple cropping index tend to have higher farmland transfer rates, and the policy facilitates the organic integration of capital with transferred farmland. This, in turn, enhances the levels of agricultural production intensification and scale, effectively strengthening food security.

## Conclusions and suggestions

Based on panel data from 1234 counties in China's major grain-producing areas spanning from 2010 to 2021, this paper conducts theoretical analysis and empirical testing on the impact of the policy of FMRML on food security. The findings indicate: Firstly, the policy significantly promotes food security in the major grain-producing areas. Robustness checks including parallel trend tests and placebo tests support this conclusion. Secondly, in terms of mechanisms, levels of mechanization and financial support partially mediate the effect of these policies, whereby enhancing rural mechanization and driving financial support can further food security in these areas. Thirdly, from the perspective of heterogeneity, the impacts of regional resource endowments, economic development levels, and farmland tenure systems on the policy of FMRML exhibit significant differences regarding food security. Notably, the policy demonstrates the most pronounced positive effects in the major grain-producing areas of the north, the eastern regions, and regions with low multiple cropping indices.

Based on the previously described findings, the following policy suggestions are presented in this article: Firstly, the policy of FMRML is crucial for promoting the long-term sustainable growth of agriculture. Lessons should be drawn from pilot regions' policy implementation experiences to further promote the adoption of this policy nationwide and globally. Secondly, accelerating the enhancement of rural financial support and mechanization levels is crucial. Financial support is pivotal in ensuring the smooth implementation of the policy of FMRML. Encouraging and supporting financial institutions to enter and expand in rural markets through initiatives like rural credit cooperatives, rural cooperative banks, or mobile financial service points would improve farmers' access to loans and financial services. Simultaneously, boosting mechanization levels directly impacts agricultural productivity and indirectly enhances food security by increasing grain production capacity. The government can

strengthen technical training and promotion efforts for agricultural machinery. Using farmland contractual management rights as collateral can alleviate financial pressures on farmers purchasing mechanical equipment. Furthermore, through technical training and on-site guidance, farmers' proficiency and efficiency in mechanized operations can be enhanced, ensuring effective grain supply and national food security. Thirdly, regional differences in resource endowments and economic bases should be considered in the farmland policy formulation process. Tailored heterogeneous FMRML policy should be developed to suit local conditions, particularly adjusting policies moderately in less economically developed regions of the southern and central-western areas to achieve coordinated regional development.

There are two limitations to this study, and it suggests topics for more investigation: First, limitations in the data have been encountered. The difficulty in collecting county-level data has resulted in incomplete data for certain variables. Specifically, while this study primarily focuses on grain yield as a measure of food security, it should be noted that the concept of food security is multidimensional and should be assessed comprehensively. Additionally, the analysis of variations in planting structures could not be conducted in depth. Second, the effectiveness of the policy is closely related to social, economic, and environmental conditions. However, this study employs a DID method to evaluate the policy effects, which does not clarify the configurational conditions under which the policy becomes effective. In the next steps, the analysis will be expanded from a configurational perspective to further examine the conditions that facilitate the effectiveness of the policy. This approach aims to provide new insights for enhancing the effectiveness of the policy of FMRML.

## Supporting information

**S1 Data. Data.**
(XLSX)

## Author contributions

**Conceptualization:** Xiangyu Dong.

**Data curation:** Xuran Li.

**Formal analysis:** Xiangyu Dong.

**Funding acquisition:** Yiru Wang.

**Investigation:** Shuiling Huang.

**Methodology:** Xiangbo Cheng.

**Resources:** Yiru Wang.

**Software:** Xuran Li.

**Supervision:** Shuiling Huang, Yiru Wang.

**Writing – original draft:** Xiangbo Cheng.

**Writing – review & editing:** Shuiling Huang.

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
