## [Decision Letter · Decision Letter 0]

16 Sep 2024

PONE-D-24-36811How Does the Farmland Management Rights Mortgage Loan Affect Food Security: Based on the Evidence of Major Grain Producing Areas in ChinaPLOS ONE

Dear Dr. Huang,

Thank you for submitting your manuscript to PLOS ONE. After careful consideration, we feel that it has merit but does not fully meet PLOS ONE’s publication criteria as it currently stands. Therefore, we invite you to submit a revised version of the manuscript that addresses the points raised during the review process.

We look forward to receiving your revised manuscript.

Kind regards,

Dingde Xu

Academic Editor

PLOS ONE

Journal Requirements:

1. When submitting your revision, we need you to address these additional requirements. Please ensure that your manuscript meets PLOS ONE's style requirements, including those for file naming. The PLOS ONE style templates can be found at https://journals.plos.org/plosone/s/file?id=wjVg/PLOSOne_formatting_sample_main_body.pdf and https://journals.plos.org/plosone/s/file?id=ba62/PLOSOne_formatting_sample_title_authors_affiliations.pdf 2. Thank you for stating the following financial disclosure: "General Research Projects of Zhejiang Provincial Department of Education".  Please state what role the funders took in the study.  If the funders had no role, please state: "The funders had no role in study design, data collection and analysis, decision to publish, or preparation of the manuscript." If this statement is not correct you must amend it as needed. Please include this amended Role of Funder statement in your cover letter; we will change the online submission form on your behalf. 3. We note that your Data Availability Statement is currently as follows: "All relevant data are within the manuscript and its Supporting Information files". Please confirm at this time whether or not your submission contains all raw data required to replicate the results of your study. Authors must share the “minimal data set” for their submission. PLOS defines the minimal data set to consist of the data required to replicate all study findings reported in the article, as well as related metadata and methods (https://journals.plos.org/plosone/s/data-availability#loc-minimal-data-set-definition). For example, authors should submit the following data: - The values behind the means, standard deviations and other measures reported;- The values used to build graphs;- The points extracted from images for analysis. Authors do not need to submit their entire data set if only a portion of the data was used in the reported study. If your submission does not contain these data, please either upload them as Supporting Information files or deposit them to a stable, public repository and provide us with the relevant URLs, DOIs, or accession numbers. For a list of recommended repositories, please see https://journals.plos.org/plosone/s/recommended-repositories. If there are ethical or legal restrictions on sharing a de-identified data set, please explain them in detail (e.g., data contain potentially sensitive information, data are owned by a third-party organization, etc.) and who has imposed them (e.g., an ethics committee). Please also provide contact information for a data access committee, ethics committee, or other institutional body to which data requests may be sent. If data are owned by a third party, please indicate how others may request data access. 4. PLOS requires an ORCID iD for the corresponding author in Editorial Manager on papers submitted after December 6th, 2016. Please ensure that you have an ORCID iD and that it is validated in Editorial Manager. To do this, go to ‘Update my Information’ (in the upper left-hand corner of the main menu), and click on the Fetch/Validate link next to the ORCID field. This will take you to the ORCID site and allow you to create a new iD or authenticate a pre-existing iD in Editorial Manager. 5. We note that Figure 1 in your submission contain [map/satellite] images which may be copyrighted. All PLOS content is published under the Creative Commons Attribution License (CC BY 4.0), which means that the manuscript, images, and Supporting Information files will be freely available online, and any third party is permitted to access, download, copy, distribute, and use these materials in any way, even commercially, with proper attribution. For these reasons, we cannot publish previously copyrighted maps or satellite images created using proprietary data, such as Google software (Google Maps, Street View, and Earth). For more information, see our copyright guidelines: http://journals.plos.org/plosone/s/licenses-and-copyright. We require you to either (1) present written permission from the copyright holder to publish these figures specifically under the CC BY 4.0 license, or (2) remove the figures from your submission: 1. You may seek permission from the original copyright holder of Figure 1 to publish the content specifically under the CC BY 4.0 license.   We recommend that you contact the original copyright holder with the Content Permission Form (http://journals.plos.org/plosone/s/file?id=7c09/content-permission-form.pdf) and the following text:“I request permission for the open-access journal PLOS ONE to publish XXX under the Creative Commons Attribution License (CCAL) CC BY 4.0 (http://creativecommons.org/licenses/by/4.0/). Please be aware that this license allows unrestricted use and distribution, even commercially, by third parties. Please reply and provide explicit written permission to publish XXX under a CC BY license and complete the attached form.” Please upload the completed Content Permission Form or other proof of granted permissions as an ""Other"" file with your submission. In the figure caption of the copyrighted figure, please include the following text: “Reprinted from [ref] under a CC BY license, with permission from [name of publisher], original copyright [original copyright year].” 2. If you are unable to obtain permission from the original copyright holder to publish these figures under the CC BY 4.0 license or if the copyright holder’s requirements are incompatible with the CC BY 4.0 license, please either i) remove the figure or ii) supply a replacement figure that complies with the CC BY 4.0 license. Please check copyright information on all replacement figures and update the figure caption with source information. If applicable, please specify in the figure caption text when a figure is similar but not identical to the original image and is therefore for illustrative purposes only.The following resources for replacing copyrighted map figures may be helpful: USGS National Map Viewer (public domain): http://viewer.nationalmap.gov/viewer/The Gateway to Astronaut Photography of Earth (public domain): http://eol.jsc.nasa.gov/sseop/clickmap/Maps at the CIA (public domain): https://www.cia.gov/library/publications/the-world-factbook/index.html and https://www.cia.gov/library/publications/cia-maps-publications/index.htmlNASA Earth Observatory (public domain): http://earthobservatory.nasa.gov/Landsat:
http://landsat.visibleearth.nasa.gov/USGS EROS (Earth Resources Observatory and Science (EROS) Center) (public domain): http://eros.usgs.gov/#Natural Earth (public domain): http://www.naturalearthdata.com/

Reviewers' comments:

Reviewer's Responses to Questions

**Comments to the Author**

1. Is the manuscript technically sound, and do the data support the conclusions?

Reviewer #1: Yes

Reviewer #2: Partly

Reviewer #3: Yes

2. Has the statistical analysis been performed appropriately and rigorously? 

Reviewer #1: Yes

Reviewer #2: Yes

Reviewer #3: Yes

3. Have the authors made all data underlying the findings in their manuscript fully available?

Reviewer #1: No

Reviewer #2: Yes

Reviewer #3: Yes

4. Is the manuscript presented in an intelligible fashion and written in standard English?

Reviewer #1: Yes

Reviewer #2: No

Reviewer #3: Yes

5. Review Comments to the Author

Reviewer #1: In this paper, through the empirical data analysis, intuitively in agricultural land management rights mortgage loan policy factors affecting food security are analyzed in detail. Food security as an important guarantee of China and the world people's life, should be comprehensive to ascend, not only on the policy to give support, but also need to invest in technology and financial resources. However, this paper also has the following is the perfect place:

First, the article lacks an in-depth and comprehensive argumentation of the farmland management right mortgage policy in the introductory part, which needs to be further refined.

Second, how does this study differ from other studies, what additions and contributions did the authors make?

Third, the addition of a theoretical mechanism diagram is suggested in section 2.2 Mechanism and Research Hypothesis of this paper.

Fourth, the article lacks some logic in its argument in the conclusion section, which should be based on the findings of this study.

Reviewer #2: The manuscript explores the impact of farmland management rights mortgage loans on food security using panel data from 1234 counties in China's major grain-producing regions from 2010 to 2021. The topic is interesting. However, there are some shortcomings as follows:

1. In the first section, the authors should address the importance and why they investigated the research topic, and some literature should also be addressed to outline the cutting-edge of this study. In the literature review part, the authors are encouraged to add the research gaps by discussing previous studies, which is pivotal in helping readers understand the paper's primary contributions. Furthermore, more updated literatures are encouraged to be added. As for the sections of introduction and literature review, the authors are encouraged to cite the following references to address the importance of this study and improve the discussions of the literature gap.

[1] Why did China's cost-reduction-oriented policies in food safety governance fail? The collective action dilemma perspective DOI: 10.1111/cjag.12313

[2] Exploring the Nexus of Enterprise Ownership Structure and Food Safety Incidents: Evidence from China DOI: 10.1080/1540496X.2023.2215891

2. For research data, the authors are recommended to update to 2022 or 2023.

3. Regarding the results of descriptive statistics, the authors should report them in general styles, incorporating minimum values and maximum values.

4. The authors are suggested to carefully proofread the manuscript before submission since there are substantial writing mistakes, typos, and the like.

Reviewer #3: This study examines the impact of farmland management rights mortgage loans (FMRML) on food security in China's major grain-producing areas. Using panel data from 1234 counties from 2010 to 2021 and employing a Difference-in-Differences model, the research finds that the FMRML policy significantly enhances food security. The following are my comments on the abstract of this paper.

Major

1. While the introduction provides background on food security and the FMRML policy in China, it doesn't clearly articulate the specific gap in the existing literature that this study aims to fill. The authors should more explicitly state what is currently unknown or understudied regarding the relationship between FMRML and food security, and how their research contributes new knowledge to this area. This would help justify the importance and novelty of the study.

2. While the introduction mentions that the study will "theoretically analyze" the impact of FMRML on food security, it doesn't provide a clear theoretical framework or hypotheses. The authors should outline the theoretical mechanisms through which they expect FMRML to affect food security, and present specific hypotheses that will be tested in the empirical analysis. This would provide a stronger foundation for the subsequent empirical work.

3. While the study considers regional differences, the analysis of policy effect heterogeneity remains superficial. I recommend further exploration of the differentiated impacts of the FMRML policy across various farmer types (e.g., smallholders vs. large-scale operators), different agricultural management models (e.g., grain cultivation vs. cash crops), and diverse land tenure structures. This would provide more targeted evidence for precise policy implementation and reveal the complex interactions between land mortgage loan policies and agricultural production structures, as well as rural social structures.

4. The conclusion should explicitly acknowledge the limitations of the current study and suggest directions for future research. This might include discussing data limitations, potential omitted variables, or alternative methodological approaches that could further validate or extend the findings. Specific areas for future investigation could be proposed, such as long-term effects of FMRML on agricultural productivity, impacts on different crop types, or effects on rural household welfare beyond food security.

Minor

1. Provide a more comprehensive and critical review of relevant literature. Clearly identify gaps in existing research and explicitly state how your study addresses these gaps. Synthesize previous findings rather than merely summarizing individual studies.

2. In the discussion section, go beyond simply restating results. Interpret your findings in the context of existing literature and broader theoretical frameworks.

6. PLOS authors have the option to publish the peer review history of their article (what does this mean? ). If published, this will include your full peer review and any attached files.

**Do you want your identity to be public for this peer review?** For information about this choice, including consent withdrawal, please see our Privacy Policy .

Reviewer #1: No

Reviewer #2: No

Reviewer #3: **Yes: ** Changgao Cheng

---

## [Author Response · Author response to Decision Letter 1]

16 Oct 2024

Response to Reviewers

Reviewer #1:

In this paper, through the empirical data analysis, intuitively in agricultural land management rights mortgage loan policy factors affecting food security are analyzed in detail. Food security as an important guarantee of China and the world people's life, should be comprehensive to ascend, not only on the policy to give support, but also need to invest in technology and financial resources. However, this paper also has the following is the perfect place:

First, the article lacks an in-depth and comprehensive argumentation of the farmland management right mortgage policy in the introductory part, which needs to be further refined.

Response 1: We thank the reviewer for pointing out this issue, the introductory section has been further justified.

Second, how does this study differ from other studies, what additions and contributions did the authors make?

Response 2: This paper offers two primary contributions. First, to the best of our knowledge, it represents the first study that utilizes extensive county-level data to analyze the interplay between FMRML policy and food security within a unified analytical framework. This investigation aims to assess whether these mortgage policies enhance food security, thereby enriching the existing literature on the impacts of the FMRML policy on food security and providing theoretical support for the findings. Second, the effects of these policies on food security can be realized through improvements in mechanization and financial support levels. Finally, considering the variations in regional natural resource endowments, economic development disparities, and differences in land tenure systems, the sample is divided into northern and southern primary grain-producing regions, eastern and central-western regions, and areas characterized by high and low double cropping indices. This allows for an exploration of the heterogeneous impacts of policy on food security.

Third, the addition of a theoretical mechanism diagram is suggested in section 2.2 Mechanism and Research Hypothesis of this paper.

Response 3: We thank the reviewer for pointing out this issue, theoretical mechanism diagrams have been added to section 2.2.

Fourth, the article lacks some logic in its argument in the conclusion section, which should be based on the findings of this study.

Response 4: We thank the reviewer for pointing out this issue, the conclusions have been further justified.

Reviewer #2:

The manuscript explores the impact of farmland management rights mortgage loans on food security using panel data from 1234 counties in China's major grain-producing regions from 2010 to 2021. The topic is interesting. However, there are some shortcomings as follows:

1. In the first section, the authors should address the importance and why they investigated the research topic, and some literature should also be addressed to outline the cutting-edge of this study. In the literature review part, the authors are encouraged to add the research gaps by discussing previous studies, which is pivotal in helping readers understand the paper's primary contributions. Furthermore, more updated literatures are encouraged to be added. As for the sections of introduction and literature review, the authors are encouraged to cite the following references to address the importance of this study and improve the discussions of the literature gap.

[1] Why did China's cost-reduction-oriented policies in food safety governance fail? The collective action dilemma perspective DOI: 10.1111/cjag.12313

[2] Exploring the Nexus of Enterprise Ownership Structure and Food Safety Incidents: Evidence from China DOI: 10.1080/1540496X.2023.2215891

Response 1: Thank you for the valuable feedback provided by the reviewers. The frontier of the relevant literature analysis has been incorporated into the first section. Additionally, the literature review has been integrated into the introduction, with the inclusion and citation of the two references mentioned above, as well as the most recent literature.

2. For research data, the authors are recommended to update to 2022 or 2023.

Response 2: Thank you for the insightful feedback provided by the reviewers. This study analyzes the implementation of the FMRML policy, which was initiated in 2016. The data utilized in this article spans from 2010 to 2021, thereby satisfying the conditions required for employing a difference-in-differences model. Furthermore, due to missing data for certain variables in 2022, the analysis is limited to data up to 2021 to ensure the rigor of the research findings.

3. Regarding the results of descriptive statistics, the authors should report them in general styles, incorporating minimum values and maximum values.

Response 3: We thank the reviewer for pointing out this issue, the descriptive statistics have been supplemented with minimum and maximum values.

4. The authors are suggested to carefully proofread the manuscript before submission since there are substantial writing mistakes, typos, and the like.

Response 4: Thank you for the valuable feedback provided by the reviewers. The manuscript has been thoroughly proofread prior to submission of the revisions.

Reviewer #3:

This study examines the impact of farmland management rights mortgage loans (FMRML) on food security in China's major grain-producing areas. Using panel data from 1234 counties from 2010 to 2021 and employing a Difference-in-Differences model, the research finds that the FMRML policy significantly enhances food security. The following are my comments on the abstract of this paper.

Major

1.While the introduction provides background on food security and the FMRML policy in China, it doesn't clearly articulate the specific gap in the existing literature that this study aims to fill. The authors should more explicitly state what is currently unknown or understudied regarding the relationship between FMRML and food security, and how their research contributes new knowledge to this area. This would help justify the importance and novelty of the study.

Response 1: Thank you for the valuable feedback provided by the reviewers. It has been recognized that the discussion of the research gap and contributions in the introduction was insufficiently clear. We will further refine this section by integrating relevant past literature to clearly identify the specific unknown areas regarding the relationship between FMRML and food security, as well as the shortcomings of existing research. The contributions and distinctive features of this study will also be articulated more clearly. Please refer to the revised introduction for these modifications.

2.While the introduction mentions that the study will "theoretically analyze" the impact of FMRML on food security, it doesn't provide a clear theoretical framework or hypotheses. The authors should outline the theoretical mechanisms through which they expect FMRML to affect food security, and present specific hypotheses that will be tested in the empirical analysis. This would provide a stronger foundation for the subsequent empirical work.

Response 2: Thank you for the valuable feedback provided by the reviewers. In the sections addressing the mechanisms and research hypotheses, a theoretical framework will be clearly constructed to detail how agricultural land operating rights mortgage loan policies impact food security, accompanied by a theoretical mechanism diagram. Additionally, several specific and testable hypotheses will be proposed based on this framework. The first section will also include further discussions of the theoretical and empirical aspects of the research.

3.While the study considers regional differences, the analysis of policy effect heterogeneity remains superficial. I recommend further exploration of the differentiated impacts of the FMRML policy across various farmer types (e.g., smallholders vs. large-scale operators), different agricultural management models (e.g., grain cultivation vs. cash crops), and diverse land tenure structures. This would provide more targeted evidence for precise policy implementation and reveal the complex interactions between land mortgage loan policies and agricultural production structures, as well as rural social structures.

Response 3: Thank you for the valuable feedback provided by the reviewers. The section on heterogeneity has been enhanced to include the impact of FMRML policy on different land tenure systems. This addition will aid readers in gaining a more comprehensive understanding of the complexities involved in the implementation of the policy.

4.The conclusion should explicitly acknowledge the limitations of the current study and suggest directions for future research. This might include discussing data limitations, potential omitted variables, or alternative methodological approaches that could further validate or extend the findings. Specific areas for future investigation could be proposed, such as long-term effects of FMRML on agricultural productivity, impacts on different crop types, or effects on rural household welfare beyond food security.

Response 4: Thank you for the valuable feedback provided by the reviewers. The limitations of this study will be discussed in detail, including data constraints and avenues for further extending the research findings. Additionally, we will actively explore future research directions, including new measurement standards and methodologies. These avenues will contribute to a more comprehensive and nuanced understanding of FMRML policy, providing policymakers with deeper insights for informed decision-making.

Revised sentence: There are two limitations to this study, and it suggests topics for more investigation: First, limitations in the data have been encountered. The difficulty in collecting county-level data has resulted in incomplete data for certain variables. Specifically, while this study primarily focuses on grain yield as a measure of food security, it should be noted that the concept of food security is multidimensional and should be assessed comprehensively. Additionally, the analysis of variations in planting structures could not be conducted in depth. Second, the effectiveness of the policy is closely related to social, economic, and environmental conditions. However, this study employs a DID method to evaluate the policy effects, which does not clarify the configurational conditions under which the policy becomes effective. In the next steps, the analysis will be expanded from a configurational perspective to further examine the conditions that facilitate the effectiveness of the policy. This approach aims to provide new insights for enhancing the effectiveness of the policy of FMRML.

Minor

1.Provide a more comprehensive and critical review of relevant literature. Clearly identify gaps in existing research and explicitly state how your study addresses these gaps. Synthesize previous findings rather than merely summarizing individual studies.

Response 1: Thank you for the valuable feedback provided by the reviewers. We will conduct a more comprehensive and systematic review of the existing literature, not only summarizing the key findings of individual studies but also focusing on integrating and comparing the perspectives, methodologies, and conclusions across different research. We will clearly identify the shortcomings present in the current literature and elaborate on how our study addresses these gaps through improvements and innovations.

2.In the discussion section, go beyond simply restating results. Interpret your findings in the context of existing literature and broader theoretical frameworks.

Response 2: Thank you for the valuable feedback provided by the reviewers. We will strive to integrate the existing literature with a broader theoretical framework to conduct a more in-depth analysis of our findings. Our goal is to present a more comprehensive, nuanced, and theoretically grounded discussion that will enhance the persuasiveness and generalizability of the research results.

---

## [Decision Letter · Decision Letter 1]

4 Nov 2024

How Does the Farmland Management Rights Mortgage Loan Affect Food Security: Based on the Evidence of Major Grain Producing Areas in China

PONE-D-24-36811R1

Dear Dr. Huang,

We’re pleased to inform you that your manuscript has been judged scientifically suitable for publication and will be formally accepted for publication once it meets all outstanding technical requirements.

Kind regards,

Dingde Xu

Academic Editor

PLOS ONE

Additional Editor Comments (optional):

Reviewers' comments:

Reviewer's Responses to Questions

**Comments to the Author**

1. If the authors have adequately addressed your comments raised in a previous round of review and you feel that this manuscript is now acceptable for publication, you may indicate that here to bypass the “Comments to the Author” section, enter your conflict of interest statement in the “Confidential to Editor” section, and submit your "Accept" recommendation.

Reviewer #1: All comments have been addressed

Reviewer #2: All comments have been addressed

2. Is the manuscript technically sound, and do the data support the conclusions?

Reviewer #1: Yes

Reviewer #2: Yes

3. Has the statistical analysis been performed appropriately and rigorously? 

Reviewer #1: N/A

Reviewer #2: Yes

4. Have the authors made all data underlying the findings in their manuscript fully available?

Reviewer #1: Yes

Reviewer #2: Yes

5. Is the manuscript presented in an intelligible fashion and written in standard English?

Reviewer #1: Yes

Reviewer #2: Yes

6. Review Comments to the Author

Reviewer #1: The paper is of a good quality so far and the editorial board could consider publishing it. After a revision, the comments of mine are addressed in a good manner.

Reviewer #2: After carefully reviewing the revised manuscript and responses, the authors have addressed most of the comments.

7. PLOS authors have the option to publish the peer review history of their article (what does this mean? ). If published, this will include your full peer review and any attached files.

**Do you want your identity to be public for this peer review?** For information about this choice, including consent withdrawal, please see our Privacy Policy .

Reviewer #1: No

Reviewer #2: No

---

## [Editor Report · Acceptance letter]

PONE-D-24-36811R1

PLOS ONE

Dear Dr. Huang,

I'm pleased to inform you that your manuscript has been deemed suitable for publication in PLOS ONE. Congratulations! Your manuscript is now being handed over to our production team.

Kind regards,

on behalf of

Dr. Dingde Xu

Academic Editor

PLOS ONE
